Research

# Prevalence of *Trichomonas vaginalis* infection and protozoan load in South African women: a cross-sectional study

Dewi J de Waaij,[1,2] Jan Henk Dubbink,[1,2] Sander Ouburg,[1] Remco P H Peters,[3,4] Servaas A Morré[1,2]

[1]Department of Medical Microbiology and Infection Control, Laboratory of Immunogenetics, VU University Medical Centre, Amsterdam, The Netherlands
[2]Department of Genetics and Cell Biology, Faculty of Health, Medicine & Life Sciences, Institute for Public Health Genomics, Research School GROW (School for Oncology & Developmental Biology), University of Maastricht, Maastricht, The Netherlands
[3]Anova Health Institute, Johannesburg and Tzaneen, South Africa
[4]Department of Medical Microbiology, University of Maastricht, Maastricht, The Netherlands

**Correspondence to**
Dr Sander Ouburg;
s.ouburg@vumc.nl

## ABSTRACT

**Objectives** *Trichomonas vaginalis* is thought to be the most common non-viral sexually transmitted infection worldwide. We investigated the prevalence, risk factors and protozoan load of *T. vaginalis* infection in South African women.

**Methods** A cross-sectional study of 604 women was conducted at 25 primary healthcare facilities in rural South Africa (Mopani district). *T. vaginalis* DNA was detected in vaginal and rectal swabs. In univariate and multivariate analyses, the *T. vaginalis* infection was investigated in relation to demographic characteristics, medical history and behavioural factors. The *T. vaginalis* load was determined as the logarithm of DNA copies per microlitre sample solution.

**Results** Collected vaginal and rectal swabs were tested for *T. vaginalis* DNA. Prevalence of vaginal *T. vaginalis* was 20% (95% CI 17.0% to 23.4%) and rectal 1.2% (95% CI 0.6% to 2.4%). Most women (66%) with a vaginal infection were asymptomatic. Factors associated with *T. vaginalis* infection were a relationship status of single (OR 2.4; 95% CI 1.5 to 4.0; p<0.001) and HIV positive infection (OR 1.6; 95% CI 1.0 to 2.6; p=0.041). Women with vaginal *T. vaginalis* infection were more likely to have concurrent *Chlamydia trachomatis* rectal infection than those without vaginal infection (12%vs3%; p<0.001; OR 4.1). A higher median *T. vaginalis* load was observed among women with observed vaginal discharge compared with those without vaginal discharge (p=0.025).

**Conclusions** Vaginal trichomoniasis is highly prevalent in rural South Africa, especially among single women and those with HIV infection, and often presents without symptoms.

## INTRODUCTION

*Trichomonas vaginalis* is a protozoan parasite that causes trichomoniasis and is mostly sexually transmitted. It is the most common non-viral sexually transmitted infection (STI) worldwide.[1] The infection is asymptomatic in 85% of the women and in 77% of the men.[1] WHO describes that *T. vaginalis* in the African region occurs 10 times as often in women than in men.[2] Vaginal *T. vaginalis* infections in the entire African region are estimated to be 42.8 million.[2] If the infection

### Strengths and limitations of this study

- ► This is the first study about the epidemiology and microbiological characteristics of *Trichomonas vaginalis* in rural South Africa.
- ► The study has a high sample size.
- ► The number of rectal *T. vaginalis* infections is relatively low.
- ► A selection bias is possible based on the geographical inclusion.

is symptomatic, women may report a change in vaginal discharge, intermenstrual bleeding or vaginal blood loss during or after sexual contact. Other symptoms that can occur are vaginal itching, dysuria and abdominal pain. Also, upper reproductive tract disease syndromes can occur, including pelvic inflammatory disease (PID). Women with a *T. vaginalis* infection have 4.7-fold increase in the risk of PID[3–5] and tubal pathology.[6] Women infected with *T. vaginalis* also have a 1.3-fold increase in the risk of preterm labour.[5]

In women, *T. vaginalis* infection can persist for months and, if left untreated, it can increase the risk for HIV acquisition if exposed.[6 7] *T. vaginalis* could lead to an increase in the vaginal HIV load and thus potentially increase the risk of HIV transmission to a sexual partner.[8] In addition, an association between *T. vaginalis* and *Chlamydia trachomatis* infection has been found.[9] Having a concurrent chlamydial infection was a predictor of both prevalent and incident *T. vaginalis*.

Despite the estimated large burden of *T. vaginalis* infection in the African region, data on clinical presentation, demographic and behavioural factors associated with infection, and microbiological factors are relatively limited. However, in other continents, multiple studies have been done in the context of *T. vaginalis* infections.[10–13] *T. vaginalis* prevalence has been reported for

a few African countries and ranges from 6.5% to 40%.[14] One of the explanations for the often high *T. vaginalis* prevalence is the lack of STI screening programmes and limited control measurements. The estimated burden of disease is significant but available data of symptomatology, coinfections and pathogen load are limited. *C. trachomatis* and *Neisseria gonorrhoeae* are other important causative micro-organisms for STI, showing an association with the pathogen load as compared with the clinical presentation and the development of long-term sequelae.[15 16]

In this study, we present the epidemiology of *T. vaginalis* infection including the *T. vaginalis* load, as measured by a validated nucleic acid amplification technique. We studied a high HIV prevalence setting in rural South Africa.

## METHODS
### Population and design
This study was part of a cross-sectional study of 604 women in rural Mopani District, South Africa, as described previously.[17] In brief, women aged 18–49 years who reported sexual activity during the prior 6 months were recruited at 25 primary healthcare facilities, regardless of the reason for visiting the facility that day. After informed consent, demographic characteristics, medical history and details of sexual behaviour were collected using a questionnaire. Healthcare workers collected vaginal and rectal swabs for molecular testing of *T. vaginalis.*

The Human Ethics Research Committee of the University of the Witwatersrand, South Africa, approved the study (Ref M110726).

### Laboratory testing
Collected vaginal and rectal swabs were analysed at the Laboratory of Medical Microbiology and Infection Control, VU University Medical Center, Amsterdam, The Netherlands. High Pure PCR Template Preparation Kit (Roche Diagnostics, Basel, Switzerland) was used for the extraction of bacterial DNA. Detection of *T. vaginalis* DNA was done with the Presto^plus assay (Microbiome, Amsterdam, The Netherlands) and the LightCycler II 480 (Roche Diagnostics) was used for the quantitative PCR for *C. trachomatis*, *N. gonorrhoeae* and *T. vaginalis.*[18] The Presto^plus assay for *C. trachomatis*, *N. gonorrhoeae* and *T. vaginalis* is currently for inhouse use only and it will be transformed into a Presto-TV assay (Goffin Molecular Technologies, The Netherlands) in 2017 in a 200 reaction format as CE-IVD certified kit. PCR positivity was determined using the crossing point (Cp) value: we used a threshold Cp of <38 for a sample to be classified as positive. Samples with Cp values of ≥38 were retested and if again ≥38 were excluded from further analyses (indeterminate samples).

### Data analyses
The following risk factors were used in our analyses: age, being single, unemployment, current pregnancy, history of vaginal discharge syndrome, currently being pregnant, the use of hormonal contraceptives, administering intra-vaginal cleansing, visiting bars, alcohol use, concurrent partners, condom use during last sex act, partner >10 years older, experiencing coercion and/or force, and sex for money or for other benefits.[17] Coercion reflects sex inequality in relationships which is associated with increased risk.[19]

Data were double entered in EpiData (Epi Info V.3.5.3) by two researchers independently, were compared for inconsistencies, cleaned and verified. The data were analysed using SPSS Statistics V.20.0 (IBM). Dichotomous data were compared using $\chi^2$ test or Fisher's exact test and continuous data using the Mann-Whitney U test. ORs with 95% CI were calculated. Vaginal *T. vaginalis* infection was defined symptomatic when either abnormal vaginal discharge during physical examination, intermenstrual bleeding or vaginal blood loss during or after sexual intercourse was reported. Univariate analysis was performed to examine possible associations of demographic variables and clinical and behavioural factors in relation to *T. vaginalis.* Multivariate analysis was conducted using a backward selection procedure and likelihood ratio tests. As a proxy for the *T. vaginalis* load, we used the median Cp value, which was calculated for each risk factor and compared between infected versus non-infected women.

## RESULTS
### Demographics of study population
A quarter of all women (n=154; 25.5%) had an age 18–24, over half of the women were single (334; 55.3%), 26 women reported anal intercourse (4.3%), a high proportion was unemployed and almost 30% reported to be infected with HIV (177; 29.3%). For more details, see table 1.

### Prevalence of *T. vaginalis* infection
Of the 604 vaginal samples, one sample was excluded based on lack of biological material and 28 gave an indeterminate result (two times Cp values of ≥38), resulting in 575 samples for analysis. For vaginal infection, 113/575 (20%) samples tested positive.

Of the 604 rectal samples, nine samples gave an indeterminate result (two times Cp values of ≥38) and were excluded resulting in 595 samples. For rectal infection, 7/595 (1.2%) samples tested positive.

Four patients were infected with *T. vaginalis* at both the rectal and vaginal sites.

### Clinical manifestation of infection
Symptoms of vaginal infection were reported by 196 (34%) of the women. Of these, 38 had *T. vaginalis* infection meaning 38/113 (34%) of the women with *T. vaginalis* reported symptoms. Of the 462 women without *T. vaginalis,* 158 reported symptoms of infection (34%). None of the individual symptoms (vaginal discharge, intermenstrual bleeding and vaginal blood loss related to

**Table 1** Univariate and multivariate analyses of factors associated with vaginal *Trichomonas vaginalis* infection in South African women

| Factor | n | % | Univariate | | Multivariate | |
|---|---|---|---|---|---|---|
| | | | OR (95% CI) | p Value | OR (95% CI) | p Value |
| Age 18–24 | 154 | 25.5 | 0.9 (0.6 to 1.6) | 0.95 | – | NS |
| Single | 334 | 55.3 | 2.4 (1.5 to 3.9) | **<0.001** | 2.4 (1.5 to 4.0) | **<0.001** |
| Unemployed | 424 | 70.2 | 1.5 (0.9 to 2.4) | 0.135 | – | NS |
| HIV infected | 177 | 29.3 | 1.6 (1.0 to 2.4) | **0.040** | 1.6 (1.0 to 2.6) | **0.041** |
| History of VDS* | 196 | 32.4 | 1.0 (0.7 to 1.6) | 0.878 | – | NS |
| Currently pregnant | 94 | 15.9 | 1.6 (1.0 to 2.7) | 0.067 | – | NS |
| Hormonal contraceptives | 230 | 33.6 | 0.6 (0.4 to 0.9) | **0.016** | – | NS |
| Intravaginal cleansing | 132 | 21.9 | 1.0 (0.6 to 1.6) | 0.841 | – | NS |
| Visits bars | 76 | 12.6 | 1.8 (1.0 to 3.1) | **0.027** | 1.7 (0.96 to 3.1) | NS |
| Alcohol use | 12 | 2 | 1.4 (0.4 to 5.1) | 0.640 | – | NS |
| Concurrent partners | 85 | 14 | 1.1 (0.7 to 1.6) | 0.799 | – | NS |
| Condom use last act | 207 | 34.3 | 1.2 (0.8 to 1.8) | 0.487 | – | NS |
| Partner >10 years older | 202 | 33.4 | 1.2 (0.8 to 1.9) | 0.335 | – | NS |
| Experienced coercion† | 169 | 28 | 0.7 (0.4 to 1.1) | 0.142 | 0.6 (0.4 to 1.0) | 0.058 |
| Experienced force | 38 | 6.3 | 1.3 (0.6 to 2.8) | 0.541 | – | NS |
| Sex for money or other benefits | 10 | 1.7 | 1.3 (0.5 to 3.3) | 0.579 | – | NS |

*Vaginal discharge syndrome (VDS) contains reported change in vaginal discharge, intermenstrual bleeding or vaginal blood loss during or after sexual contact.
†Coercion reflects sex inequality in relationships, associated with increased risk.[17]

sexual contact) was associated with *T. vaginalis* infection. For rectal infection (n=7), the numbers were low and no rectal symptomatology was scored.

### Factors associated with *T. vaginalis* infection

In univariate analysis, we observed significant associations for single status, being HIV positive, the use of hormonal contraceptives and visiting bars (table 1). In multivariate analysis (table 1), having a relationship status that is single was significantly associated with a vaginal *T. vaginalis* infection (adjusted OR (aOR) 2.4; 95% CI 1.5 to 4.0; p<0.001) and the presence of HIV infection was significantly associated (aOR 1.6; 95% CI 1.0 to 2.6; p=0.041) *T. vaginalis* infection. Within women with a current vaginal *T. vaginalis* infection, a rectal *C. trachomatis* infection is significantly more present (p<0.001, OR 4.1).

### Determinants of *T. vaginalis* load

The median Cp value for vaginal *T. vaginalis* load of infection was 28.0 (range 17.0–37.9). We did not observe any significant associations between the *T. vaginalis* load and demographic factors (table 2). Also, no association was found between the *T. vaginalis* load and symptoms in general. However, in the group of women with *T. vaginalis* infection who had vaginal discharge observed during examination, a significantly higher load was observed as compared with those without vaginal discharge (p=0.025).

### DISCUSSION

In this study, we describe the prevalence, risk factors and the load of *T. vaginalis* infection. This is one of a few studies to report these aspects for *T. vaginalis* in a cohort from South Africa. We observed a vaginal *T. vaginalis* prevalence of 20% (95% CI 17.0% to 23.4%) and a rectal prevalence of 1.2% (95% CI 0.6% to 2.4%).

In multivariate analysis, two risk factors for vaginal *T. vaginalis* infection were identified: having a relationship status that is single and the presence of HIV infection. In addition, within women with vaginal *T. vaginalis* infection, a rectal *C. trachomatis* infection is significantly more present. Finally, a high *T. vaginalis* load appears to be associated with vaginal discharge observed during examination.

The prevalence of vaginal *T. vaginalis* infection in our study corresponds with the range reported by other studies in Africa (3.2%–40%).[14 20] Rectal *T. vaginalis* infections are rarely described. One study from San Francisco, USA, describes a rectal *T. vaginalis* prevalence of 0.6%, based on 'men who have sex with men' who reported receptive anal sex within 6 months prior to their clinic visit.[21 22]

We show two demographic factors that may contribute to the high vaginal *T. vaginalis* prevalence in our study. First, we observed that a *T. vaginalis* infection is associated with a single relationship status. Joffe *et al* described that a single marital status is often associated with having more

| Risk factor | Median Cp | p Value |
|---|---|---|
| **Age 18–24** | | |
| 18–24 | 28 | 0.747 |
| >24 | 27.7 | |
| **Single** | | |
| Yes | 27.3 | 0.971 |
| No | 28 | |
| **Unemployed** | | |
| Yes | 27.7 | 0.810 |
| No | 29.7 | |
| **HIV infected** | | |
| Yes | 27.5 | 0.986 |
| No | 27.8 | |
| **History of VDS** | | |
| Yes | 28 | 0.954 |
| No | 27.6 | |
| **Currently pregnant** | | |
| Yes | 28 | 0.794 |
| No | 27.7 | |
| **Hormonal contraceptives** | | |
| Yes | 27.6 | 0.464 |
| No | 28 | |
| **Intravaginal cleansing** | | |
| Yes | 28.5 | 0.433 |
| No | 27.7 | |
| **Visits bars** | | |
| Yes | 25.3 | 0.310 |
| No | 28 | |
| **Alcohol use** | | |
| Yes | 27.9 | 0.613 |
| No | 30.2 | |
| **Concurrent partners** | | |
| Yes | 27 | 0.335 |
| No | 28 | |
| **Condom use last sex act** | | |
| Yes | 27 | 0.339 |
| No | 28 | |
| **Experiences force** | | |
| Yes | 29 | 0.603 |
| No | 27.9 | |

**Table 2** *Trichomonas vaginalis* load in association with risk factors

VDS, vaginal discharge syndrome

than one sexual partner and thereby an increased risk for STI in general.[23] To be more specific, it has been demonstrated that women having more than two concurrent male sexual partners are more likely to have a *T. vaginalis* infection.[24] Although we were not able to confirm this, we hypothesise that a single marital status is associated with having more than one concurrent sexual partner.

It has been suggested that *T. vaginalis* infection is age tied.[25 26] Women, who were older than 20 years of age, were more likely to be infected with *T. vaginalis*.[27] In our study, we did not observe an association between age and *T. vaginalis* infections.

We confirm an association of positive HIV status with presence of *T. vaginalis* infection, which has been reported by several other studies.[4 8 28–31] *T. vaginalis* infection was present in almost a quarter of HIV-infected women in our study. This is in line with the previously described higher *Trichomonas* prevalence in high HIV prevalence cities compared with low HIV prevalence cities.[14 20] The biological link between HIV and TV is yet unclear; however, bacterial vaginosis is described to be an independent risk factor for acquisition of STI, and acquisition and transmission of HIV.[32] Although we did not include bacterial vaginosis, it is described to be associated with a 1.5-fold to 2-fold increased risk for incident trichomonal infection.[33] This could suggest an association between HIV and *T. vaginalis* where an asymptomatic *T. vaginalis* infection could increase the risk of HIV infection.

It has been described that non-sexual transmission of *T. vaginalis* occurs in girls who did not have sexual intercourse.[34 35] A striking prevalence of 43% was found in a study among women from Ndola, Zambia, who denied ever had sexual intercourse.[14] In those cases, *T. vaginalis* would be transmitted by wet clothes and water, since some studies suggest that *T. vaginalis* can survive up to days in warm mineral water and also for hours in urine.[34] *T. vaginalis* is described to have some kind of robustness, which may explain its different transmission ways compared with other STIs. This may be an explanation of many hypotheses for the high prevalence; however, they need to be further explored.

Twenty-six per cent of all women reported symptoms, without being diagnosed with a *T. vaginalis* infection. Although 20% of these women were diagnosed with another vaginal STI than *T. vaginalis*, symptoms are evidently a poor predictor for a *T. vaginalis* infection. We did not find a significant difference in symptomatology and *T. vaginalis* infection status. More knowledge about *T. vaginalis* disease would provide valuable information for potential targeted treatment approaches versus syndromic management.

In general, self-reported symptoms are not reliable in predicting an STI, like *T. vaginalis*.[36 37] However, 31 (16%) of the women vaginally infected with *T. vaginalis* were coinfected with either *C. trachomatis* or *N. gonorrhoeae* (data not shown). Symptoms of these frequent occurring STI may show comparable symptoms and should be taken into consideration. Although other studies revealed significant rates for coinfection between *T. vaginalis* and other STIs,[38] in the current study we did not observe an increased risk for coinfection with *C. trachomatis* and/or *N. gonorrhoeae* in patients infected with *T. vaginalis* (data not shown). However, in a previous study from our group,

we describe an increased risk of obtaining a *T. vaginalis* coinfection when infected with either *C. trachomatis* or *N. gonorrhoeae*. Concurrent STIs may be due to common risk factors for these infections, such as sexual activity, contraceptive use and new partners.

This is one of a few studies to report aspects of rectal *T. vaginalis* in a cohort from South Africa. We observed that rectal *C. trachomatis* infections are significantly more present in combination with a vaginal *T. vaginalis* infection. This may be due to several factors. First, sexual behaviour including anal sex is an important factor that may play a role in the presence of coinfections. Second, there is the ability to get reinfected via the gastrointestinal tract by autoinoculation, after being cured of the vaginal infection for *Chlamydia*, but it is unclear if this route of infection is also present for *T. vaginalis* infections and thus if the rectal site could be the reservoir for *T. vaginalis* infection.[39] These potential transmission routes may result from vaginal wiping techniques, poor personal hygiene, using fingers during sex or for cleansing afterwards and use of sex toys.[17] The most likely explanation is that the rectal samples may be contaminated by cervicovaginal fluid due to the close anatomical proximity[40 41] and that *T. vaginalis* positive rectal samples in women were contaminated with *T. vaginalis* positive vaginal fluid of the perianal surface.[42]

It is conceivable that micro-organism load may affect the clinical course of infection. It has been shown in studies of another STI, *C. trachomatis*, that there is an association between the bacterial load and genital manifestations of infection.[43 44] A higher *C. trachomatis* bacterial load is associated with the presence of at least two specific symptoms (vaginal discharge, irregular bleeding, or pelvic or abdominal pain). The impact of *T. vaginalis* load has not yet been described. We hypothesised that *T. vaginalis* load may affect the clinical course of the infection and compared several factors with the load of *T. vaginalis*. We did not find any significant association between the load and reported symptoms of a *T. vaginalis* infection. However, within women with observed vaginal discharge during examination, a significantly higher load was found. The load determination was performed using Cp values of the *T. vaginalis* PCR. Another technique to determine the average load per human is to incorporate the number of human cells. Our results prove that there is an association between *T. vaginalis* load and discharge. However, the literature on *T. vaginalis* load is scarce, and the effect of the load has to be further explored.

Although this is a unique study about the epidemiology and microbiological characteristics of *T. vaginalis* in rural South Africa with a relatively good sample size, a limitation of our study is that the number of rectal *T. vaginalis* infections is rather small for further analyses. As a consequence, this could have played a role in our statistical analyses. In addition, there can be a selection bias based on for instance the geographical inclusion, and self-reported symptoms can be more subjective than clinician-based symptomatology. We cannot exclude potential cross-contamination between the cervicovaginal and anorectal sites. The results of this may be generalised to similar populations; however, these results might not be extended to other populations due to the aforementioned potential geographical and reporting biases.

In conclusion, *T. vaginalis* infections are highly prevalent in the rural South African population investigated. The asymptomatic nature, potential for non-sexual transmission especially in socioeconomic poor settings and the syndromic management approach to STI control in this region are potential factors contributing to the prevalence. We observed a low rectal prevalence, part of which might be contamination with *T. vaginalis* positive vaginal fluid of the perianal surface. In multivariate analyses being single or HIV positive was identified as a risk factor for vaginal *T. vaginalis* infection. Finally, a high *T. vaginalis* load seems to be associated with vaginal discharge observed during examination.

**Contributors** DJdW: laboratory analyses, statistical analyses and writing of the manuscript. JHD: sample collection and writing of the manuscript. SO: statistical analyses, reviewing of the manuscript and study supervision. RPHP: sample collection, reviewing of the manuscript and study supervision. SAM: reviewing of the manuscript and study supervision.

**Funding** This work was supported by the Dutch Society for Tropical Medicine (NVTG), the Netherlands. The Anova Health Institute is supported by the US President's Emergency Plan for AIDS Relief programme via the US Agency for International Development under Cooperative Agreement Number AID-674-A-12-00015.

**Disclaimer** The views expressed in this manuscript do not necessarily reflect those of the President's Emergency Plan for AIDS Relief or the US Agency for International Development.

**Competing interests** SAM is a full-time employee of the VU University Medical Center Amsterdam, and has been involved in the technical development of the Presto[plus] CT-NG-TV assay (Microbiome, The Netherlands) via Microbiome Ltd (SAM is co-founder and co-director), a spin-off company of the VU University Medical Center, Amsterdam, The Netherlands. None of the other authors report a potential conflict of interest.

**Patient consent** Obtained.

**Ethics approval** Human Ethics Research Committee of the University of the Witwatersrand, South Africa.

**Provenance and peer review** Not commissioned; externally peer reviewed.

**Data sharing statement** All data of this study have been described in the paper. No additional data for this study are available.

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
