## [Reviewer comments · BMJ Open]

ARTICLE DETAILS

TITLE (PROVISIONAL)	Prevalence of Trichomonas vaginalis infection and protozoan load in South African women: a cross-sectional study
AUTHORS	de Waaij, Dewi; Dubbink, Jan Henk; Ouburg, Sander ; Peters, Remco; Morre, S

VERSION 1 – REVIEW

REVIEWER	Dr Melanie. Bissessor Melbourne Sexual Health Centre Australia
REVIEW RETURNED	12-Apr-2017

GENERAL COMMENTS	Thank you for a well written manuscript The research question is novel and interesting Abstract: satisfactory and acceptable Introduction This is well referenced and introduces the study question very well. Methods: There is some repetition under data analyses which needs to be removed Results: the age range of the cohort needs to be indicated, not just those women less than 24 years Tables are very clear with odds ratios and p values The load data is also well presented Discussion Lines 41 to 43 need to be revised. The authors did observe an association with age but not in the same direction as other studies. These studies need to be referenced. Lines 56 to 57: the authors need to indicate that bacterial vaginitis was not measured in this study and therefore cannot be compared to the study referenced as 23. Lines 28 to 39 of page 11: can the authors suggest a reason for a lack of co-infection risk with TV and chlamydia and gonorrhoea? In the limitations section - lines 27 to 35, cross contamination of rectal and vaginal swabs needs to be discussed again. The conclusion is well written. Well done on a great study and well written manuscript.
--

REVIEWER	Erik Munson Marquette University USA Within the past year, I have received travel assistance from Hologic, Incorporated
REVIEW RETURNED	05-May-2017

GENERAL COMMENTS	In the manuscript "Prevalence of Trichomonas vaginalis infection and protozoan load in South African women" (bmjopen-2017-016959), de Waaij et al. use a DNA-based molecular assay to characterize T. vaginalis prevalence and nucleic acid burden in a rural South African female cohort. While a 1.2% detection rate was noted from rectal samplings of all women, an approximate 20% detection rate emanated from vaginal swab samples. Symptomatic patients were more likely to have higher T. vaginalis burden (as measured by crossing point values of the molecular assay). In multivariate analyses, marital status (single) and infection with human immunodeficiency virus (HIV) were associated with T. vaginalis detection. In general, the manuscript was fairly well written, with presence of grammatical errors (including issues with subject/verb agreement). The major concern from this Reviewer revolves around novelty of findings/noteworthy contribution to the literature. In addition, this Reviewer is cognizant that trichomoniasis prevalence may vary in different regions of the world; with that said, many of the assertions made in the Discussion may not be completely accurate. Specific comments follow:  1) With respect to the Introduction sentence, "Despite the estimated large burden...data on clinical presentation, demographic, and behavioural factors associated with infection, and microbiological factors is relatively limited", this Reviewer strongly disagrees. One is referred to multiple papers published by the laboratories of Charlotte Gaydos, Marcia Hobbs, Arlene Sena, Jane Schwebke, and others; 2) This Reviewer is uncertain with the means that the authors took to determine whether a final result was indeterminate. Without seeing other information in the Methods section, it appears as if this were ascertained from nebulous crossing point values (even upon repeat testing) of target T. vaginalis amplicon. There is no mention of the presence of an internal control nucleic acid sequence in assay master mix (or detection thereof). This would be a more meaningful means of assessing for validity of results from clinical specimens because it factors in the potential for inhibitors of nucleic acid amplification; 3) "Use of hormonal contraceptives" appears to be listed twice within the data analyses section; 4) This Reviewer is having difficulty understanding why rectal sampling of patients would occur, as only a 1.2% detection rate was noted (and of those positive patients, over one-half were detected by a concomitant vaginal swab). Are the authors proponents of this double-sampling method on a routine basis? If so, what impact would this have on clinical testing laboratories (especially with respect to expenditures related to testing)? 5) With respect to the fourth paragraph of the Discussion section, many data exist with respect to female age and T. vaginalis detection. The authors are referred to publications from the aforementioned authors, as well as papers published by the laboratories of Kimberle Chapin and Erik Munson.
--

	Many of these publications describe an association between T. vaginalis and older individuals; however studies in high-prevalence STI locations also show high prevalence rates in young females-- particularly when RNA amplification methods are utilized; 6) With respect to the fifth paragraph of the Discussion section, many data describe a link between T. vaginalis and HIV. The authors are referred to papers published by the laboratories of Patricia Kissinger and John Alderete; 7) The authors noted that a significant percentage of symptomatic patients was not diagnosed with T. vaginalis infection. Could this be related to assay sensitivity? The authors are referred to the paper published by Melinda Nye in American Journal of Obstetrics and Gynecology which discussed differences in analytic sensitivity of RNA amplification versus those of DNA amplification; 8) Later in that paragraph, the authors opine that more knowledge about T. vaginalis disease would provide data for targeted treatment approaches versus syndromic management. With all due respect, this Reviewer disagrees; there appears to be too much symptom overlap between trichomoniasis and other STI to make this possible; 9) It should be noted that several studies have revealed significant co-detection rates of T. vaginalis with Chlamydia trachomatis and/or Neisseria gonorrhoeae. The authors are referred to studies from the laboratory of Jill Huppert and others; 10) With respect to non-sexual transmission of T. vaginalis, the authors may consider citing studies that are more contemporary than reference 25 (1984); and, 11) The association of increased organism burden with symptomatic trichomoniasis is not surprising to this Reviewer. Nucleic acid hybridization assays specific for T. vaginalis (with less sensitivity when compared to nucleic acid amplification testing) have greatest utility in symptomatic trichomoniasis. In contrast, nucleic acid amplification assays can be applied to both symptomatic and asymptomatic patients. This comparison also infers that symptomatic trichomoniasis is related to organism burden.
--	---

VERSION 1 – AUTHOR RESPONSE

The comments of Reviewer 1:

Thank you for the valuable comments and your praising words. Below you will find the responses on the specific comments.

Abstract: satisfactory and acceptable

Introduction

This is well referenced and introduces the study question very well.

Methods:

Comment: There is some repetition under data analyses which needs to be removed

Response: We deleted the duplication of the 'use of hormonal contraceptives'

Results:

Comment: the age range of the cohort needs to be indicated, not just those women less than 24 years

Response: We changed ≤ 24 into 18-24

Comment: Tables are very clear with odds ratios and p values
The load data is also well presented

Discussion

Comment: Lines 41 to 43 need to be revised. The authors did observe an association with age but not in the same direction as other studies. These studies need to be referenced.

Response: We did not find an association between *T. vaginalis* infections and age. However, we refer to studies that did find associations between *T. vaginalis* infections and age.

Comment: Lines 56 to 57: the authors need to indicate that bacterial vaginitis was not measured in this study and therefore cannot be compared to the study referenced as 23.

Response: We have added that we did not include data of bacterial vaginosis

Comment: Lines 28 to 39 of page 11: can the authors suggest a reason for a lack of co-infection risk with TV and chlamydia and gonorrhoea?

Response: Because *T. vaginalis* and *C. trachomatis* are both common STIs, we expected to find more cases with co-infections. However, we did not. In another study from our group, we found that having either *C. trachomatis* or *N.gonorrhoeae* increases the risk for a *T. vaginalis* infection.

Comment: In the limitations section - lines 27 to 35, cross contamination of rectal and vaginal swabs needs to be discussed again.

Response: We conclude that prevalence rates may be influenced by spillover and contamination

Comment: The conclusion is well written.

Response: Well done on a great study and well written manuscript.

The comments of Reviewer 2:

Thank you for the valuable comments. Below we have responded to the specific comments.

1) With respect to the Introduction sentence, "Despite the estimated large burden...data on clinical presentation, demographic, and behavioural factors associated with infection, and microbiological factors is relatively limited", this Reviewer strongly disagrees. One is referred to multiple papers published by the laboratories of Charlotte Gaydos, Marcia Hobbs, Arlene Sena, Jane Schwebke, and others;

Response: We added references and adapted the sentence

2) This Reviewer is uncertain with the means that the authors took to determine whether a final result was indeterminate. Without seeing other information in the Methods section, it appears as if this were ascertained from nebulous crossing point values (even upon repeat testing) of target *T. vaginalis* amplicon. There is no mention of the presence of an internal control nucleic acid sequence in assay master mix (or detection thereof). This would be a more meaningful means of assessing for validity of results from clinical specimens because it factors in the potential for inhibitors of nucleic acid amplification;

Response: The threshold of <38 and repeated testing is quite normal in routine diagnostics. The definition of indeterminate too. In normal routine diagnostics after the indeterminate status has been assessed one requests a new independent sample from the patient. This is, however, not feasible in the current study in South Africa.

We have unfortunately no internal control results to monitor inhibition. The potential effect of inhibition could have resulted in a slightly underestimated TV prevalence but based on the already high TV prevalence and the on average small inhibition percentages in current diagnostics, based on good DNA extraction and robust PCR, only a limited effect on missed TV positive cases affecting overall TV prevalence is expected. We have made a remark on this in the discussion. Finally, the clinical association found between load and symptoms will not be affected by potential inhibition since the effect will most likely be distributed equally among TV positive and negative women and will result in the same small increase in strong, intermediate, and low positive TV cases.

3) "Use of hormonal contraceptives" appears to be listed twice within the data analyses section;.

Response: We deleted the duplication

4) This Reviewer is having difficulty understanding why rectal sampling of patients would occur, as only a 1.2% detection rate was noted (and of those positive patients, over one-half were detected by a concomitant vaginal swab). Are the authors proponents of this double-sampling method on a routine basis? If so, what impact would this have on clinical testing laboratories (especially with respect to expenditures related to testing)?

Response: A double-sampling method would have an impact on the laboratory that more assays are needed with the additional costs. However, a study from van Lier et al.* found that a proportion of anorectal-only chlamydia infections was substantial: 38% (n = 11) in women with indication (reported anal sex or symptoms) and 19% (n = 19) in women without indication. This concludes that several cases could be missed and anorectal testing is indicated.

* High Proportion of Anorectal Chlamydia trachomatis and Neisseria gonorrhoeae After Routine Universal Urogenital and Anorectal Screening in Women Visiting the Sexually Transmitted Infection Clinic, Clin Infect Dis 2017

5) With respect to the fourth paragraph of the Discussion section, many data exist with respect to female age and T. vaginalis detection. The authors are referred to publications from the aforementioned authors, as well as papers published by the laboratories of Kimberle Chapin and Erik Munson. Many of these publications describe an association between T. vaginalis and older individuals; however studies in high-prevalence STI locations also show high prevalence rates in young females--particularly when RNA amplification methods are utilized;

Response: We added references mentioned above

6) With respect to the fifth paragraph of the Discussion section, many data describe a link between T. vaginalis and HIV. The authors are referred to papers published by the laboratories of Patricia Kissinger and John Alderete;

Response : We added references mentioned above

7) The authors noted that a significant percentage of symptomatic patients was not diagnosed with *T. vaginalis* infection. Could this be related to assay sensitivity? The authors are referred to the paper published by Melinda Nye in American Journal of Obstetrics and Gynecology which discussed differences in analytic sensitivity of RNA amplification versus those of DNA amplification;

Response: The paper mentioned above describes that APTIMA *Trichomonas vaginalis* transcription-mediated amplification is more sensitive than wet mount or culture in vaginal swabs. In males, the ATV TMA was significantly more sensitive than culture or PCR. However, we used the PrestoPlus assay, what our group validated recently, with a sensitivity of 95%. #

Evaluation of Prestoplus assay and LightMix kit *Trichomonas vaginalis* assay for detection of *Trichomonas vaginalis* in dry vaginal swabs, de Waaij et al., J Microbiol Methods 2016

8) Later in that paragraph, the authors opine that more knowledge about *T. vaginalis* disease would provide data for targeted treatment approaches versus syndromic management. With all due respect, this Reviewer disagrees; there appears to be too much symptom overlap between trichomoniasis and other STI to make this possible;

Response: We agree, however, syndromic management includes triple therapy for three STI's: Azithromycin for *C. trachomatis*, Ceftriaxone for *N. gonorrhoeae*, and Metronidazole for *T. vaginalis*. More data for targeted treatment approach would include specific therapy for an STI without the risk for possible resistance for antibiotics. Furthermore, more data may help develop strategies to help identify at risk asymptomatic patients that would be missed by the syndromic approach and are thus left untreated.

9) It should be noted that several studies have revealed significant co-detection rates of *T. vaginalis* with *Chlamydia trachomatis* and/or *Neisseria gonorrhoeae*. The authors are referred to studies from the laboratory of Jill Huppert and others;

Response: We added the reference mentioned above

10) With respect to non-sexual transmission of *T. vaginalis*, the authors may consider citing studies that are more contemporary than reference 25 (1984);

Response: We added a more current reference

11) The association of increased organism burden with symptomatic trichomoniasis is not surprising to this Reviewer. Nucleic acid hybridization assays specific for *T. vaginalis* (with less sensitivity when compared to nucleic acid amplification testing) have greatest utility in symptomatic trichomoniasis. In contrast, nucleic acid amplification assays can be applied to both symptomatic and asymptomatic patients. This comparison also infers that symptomatic trichomoniasis is related to organism burden.

Response: Burden of a microorganism does not necessarily correlate to a symptomatic course of disease. For instance, some *C. trachomatis* infected asymptomatic women have been observed to have higher loads compared to *C. trachomatis* infected symptomatic women.

The reviewer may not find it surprising, however in this study we show with observational data that *T. vaginalis* load is correlated to a symptomatic course of infection.

VERSION 2 – REVIEW

REVIEWER	Melanie Bissessor Melbourne Sexual Health Centre Australia
REVIEW RETURNED	18-Jun-2017

GENERAL COMMENTS	This a well written manuscript with the outcomes clearly defined. The methodology is comprehensive, including the statistical review. The authors have answered the queries in the initial review.
--

REVIEWER	Erik Munson Marquette University USA
REVIEW RETURNED	03-Jul-2017

GENERAL COMMENTS	In the revised manuscript “Prevalence of Trichomonas vaginalis infection and protozoan load in South African women” (bmjopen-2017-016959.R1), de Waaij et al. use a DNA-based molecular assay to characterize T. vaginalis prevalence and nucleic acid burden in a rural South African female cohort. While a 1.2% detection rate was noted from rectal samplings of all women, an approximate 20% detection rate emanated from vaginal swab samples. Symptomatic patients were more likely to have higher T. vaginalis burden (as measured by crossing point values of the molecular assay). In multivariate analyses, marital status (single) and infection with human immunodeficiency virus (HIV) were associated with T. vaginalis detection. In general, the manuscript was well written and many of the initial concerns of the two reviewers were satisfied. References were updated in satisfactory fashion. The major concern from this Reviewer continues to revolve around novelty of findings/noteworthy contribution to the literature. This Reviewer continues to question the necessity of screening females for T. vaginalis infection via rectal collections. Specific comments follow: 1) If only 4% of females claim to be involved in anal intercourse, why were rectal specimens collected from all study participants? How many of the positive rectal T. vaginalis PCR results were generated from females who did not claim anal intercourse?2) If positive rectal T. vaginalis PCR results were derived from women not claiming to engage in anal sexual practices, is this a clinically-significant finding? Alternatively, would this be indicative of cross-contamination from genital sites (to which the authors allude in the conclusion paragraph)? Could these positive rectal T. vaginalis PCR findings be confirmed by alternative PCR assay (different primers, different manufacturers)?3) The authors do cite one paper about T. vaginalis nucleic acid detection from rectal specimens (though I think that reference 21 may be an incorrect reference), but this was a study done in males. I feel that the authors need to perform an extensive literature review on this paradigm in female populations;4) Is the Prestoplus assay CE marked (or have other registration)?5) This Reviewer continues to feel that the assay could benefit from a check on endogenous inhibitors of nucleic acid amplification. Optimally, this should be done through inclusion of an internal control sequence within the assay.
--

	The “indeterminate” rate of approximately 5% from vaginal specimens seems rather high; and, 6) The association of increased organism burden with symptomatic trichomoniasis is not surprising to this Reviewer and may not be a novel finding.
--	--

VERSION 2 – AUTHOR RESPONSE

Responses to the reviewers.

We would like to thank the reviewers for their comments and suggestions. We address the specific points below.

Reviewer 2 questions:

1) If only 4% of females claim to be involved in anal intercourse, why were rectal specimens collected from all study participants? How many of the positive rectal *T. vaginalis* PCR results were generated from females who did not claim anal intercourse?

Response: Rectal specimens were collected due to plausible auto-inoculation and to investigate the rectal prevalence in general in this population. There was no relation between reported anal intercourse and TV positivity, actually none of the women who reported anal intercourse was rectally TV positive.

2) If positive rectal *T. vaginalis* PCR results were derived from women not claiming to engage in anal sexual practices, is this a clinically-significant finding? Alternatively, would this be indicative of cross-contamination from genital sites (to which the authors allude in the conclusion paragraph)? Could these positive rectal *T. vaginalis* PCR findings be confirmed by alternative PCR assay (different primers, different manufacturers)?

Response: Yes, we indeed state in the discussion that cross contamination or autoinoculation is the most logical explanation for these rectal positive results. In theory we could use another TV assay to confirm our findings. However, we have not done so since we already made a technical comparison of our TV PRESTO Assay to the LightMix from TibMolBiol with Qiagen for the discrepancy analyses, so we know the Assay is reliable since no significant differences in performances were discovered. We also referred to this assay in our article (reference no. 18)

Article:

Evaluation of Presto(plus) assay and LightMix kit *Trichomonas vaginalis* assay for detection of *Trichomonas vaginalis* in dry vaginal swabs, de Waaij DJ, Ouburg S, Dubbink JH, Peters RP, Morré SA., *J Microbiol Methods*. 2016 Aug;127:102-4.

3) The authors do cite one paper about *T. vaginalis* nucleic acid detection from rectal specimens (though I think that reference 21 may be an incorrect reference), but this was a study done in males. I feel that the authors need to perform an extensive literature review on this paradigm in female populations;

Response: In the Netherlands and also outside the Netherlands rectal specimens are used often for STD detection, especially amongst females who are rectally positive for CT also to detect LGV, specifically LGV L2b. In the Netherlands we also identified the first women in the world with the MSM L2b strain rectally. These rectal specimens are thus used often in females and in males (especially MSM) and no differences are reported between male and female rectal specimens with regard to DNA isolation efficiency. I have replaced our reference based on male rectal samples by two representative female rectal specimen studies from our own group to show the expertise we have.

Reference 21:

Analyses of multiple-site and concurrent Chlamydia trachomatis serovar infections, and serovar tissue tropism for urogenital versus rectal specimens in male and female patients., Bax CJ, Quint KD, Peters RP, Ouburg S, Oostvogel PM, Mutsaers JA, Dörr PJ, Schmidt S, Jansen C, van Leeuwen AP, Quint WG, Trimbos JB, Meijer CJ, Morré SA., Sex Transm Infect. 2011 Oct;87(6):503-7.

Reference 22:

The first case record of a female patient with bubonic lymphogranuloma venereum (LGV), serovariant L2b., Verweij SP, Ouburg S, de Vries H, Morré SA, van Ginkel CJ, Bos H, Sebens FW., Sex Transm Infect. 2012 Aug;88(5):346-7.

4) Is the Prestoplus assay CE marked (or have other registration)?

Response: The Presto Assay is CE IVD marked for CT and NG, the Prestoplus CT NG TV is for in house use only and the Prestoplus will be available this year in a 200 reaction format as CE-IVD certified Kit.

5) This Reviewer continues to feel that the assay could benefit from a check on endogenous inhibitors of nucleic acid amplification. Optimally, this should be done through inclusion of an internal control sequence within the assay. The "indeterminate" rate of approximately 5% from vaginal specimens seems rather high; and,

Response: We agree, but at this moment these data is not available nor possible to make. However, we check from human DNA by a HLA target and know this target is positive. Though is does not exclude partial or limited inhibition, we are confident the inhibition does not play a major role also since the STD prevalence for TV and in the past for CT and NG and MG are rather high showing no major inhibition problem.

6) The association of increased organism burden with symptomatic trichomoniasis is not surprising to this Reviewer and may not be a novel finding.

Response: We agree with the reviewer it is not a surprising finding since it is also linked to sexual behavior. It is however not often reported in South African women yet, and relevant also from a policy perspective to change attitudes and initiate screening modalities